# Construction of C-B axial chirality via dynamic kinetic asymmetric cross-coupling mediated by tetracoordinate boron

Kai Yang[1,4], Yanfei Mao[1,4], Zhihan Zhang[2,4], Jie Xu[1], Hao Wang[1], Yong He[1], Peiyuan Yu [2] ✉ & Qiuling Song [1,3] ✉

Catalytic dynamic kinetic asymmetric transformation (DyKAT) provides a powerful tool to access chiral stereoisomers from racemic substrates. Such transformation has been widely employed on the construction of central chirality, however, the application in axial chirality remains underexplored because its equilibrium of substrate enantiomers is limited to five-membered metalacyclic intermediate. Here we report a tetracoordinate boron-directed dynamic kinetic asymmetric cross-coupling of racemic, configurationally stable 3-bromo-2,1-azaborines with boronic acid derivatives. A series of challenging C-B axially chiral compounds were prepared with generally good to excellent enantioselectivities. Moreover, this transformation can also be extended to prepare atropisomers bearing adjacent C-B and C-C diaxes with excellent diastereo- and enantio-control. The key to the success relies on the rational design of a reversible tetracoordinate boron intermediate, which is supported by theoretical calculations that dramatically reduces the rotational barrier of the original C-B axis and achieves the goal of DyKAT.

Catalytic dynamic kinetic asymmetric transformation (DyKAT) has emerged as a powerful platform for 100% theoretical conversion of racemic, configurationally stable substrates into high-value optically pure compounds[1,2] like numerous pharmaceuticals and natural products[3–6]. Mechanistically, these reactions normally entail a chiral catalyst-mediated equilibration of substrate enantiomers, involving formation of diastereomeric substrate–catalyst intermediates with unstable configuration (DyKAT type I, Fig. 1a) or a common chiral intermediate that has lost the substrate's chiral center (DyKAT type II, Fig. 1a). DyKAT strategy has been widely employed on central chirality[2], in recent years, it has also demonstrated important applications on the construction of heterobiaryl atropisomers[7], which are prevalent in natural products, medicines, ligands, catalysts and materials[8–17]. The transition-metal-catalyzed DyKAT of racemic, configurationally stable heterobiaryl substrates have been applied in the

synthesis of axially chiral heterobiaryl compounds[18–25] since the pioneering works by Lassaletta & Fernandez[18] and Virgil & Stoltz[19]. The key of these elegant works all depends on the fast interconversion of diastereomeric substrate–catalyst intermediates promoted by five-membered metalacyclic intermediates[18–25] (Fig. 1b). However, the monotonous reaction mode significantly restricts the wide application of DyKAT on axial chirality, and the challenge is to explore and find more modes to promote equilibrium of substrate enantiomers.

Compared with the common $C(sp^2)$–$C(sp^2)$ atropisomers, the $C(sp^2)$–$C(sp^3)$ atropisomers have a lower rotational barrier because the conical space of $sp^3$ carbon is more conducive to rotation[26–33] (Fig. 1c). It can be imagined that (1) if an atom of the stereogenic axis of the diastereomeric intermediates in DyKAT changes from $sp^2$ to $sp^3$ under the action of additional reagents, the group on the stereogenic axis may be easier to rotate; (2) if the conversion from $sp^2$ to $sp^3$ is

[1]Key Laboratory of Molecule Synthesis and Function Discovery, Fujian Province University, College of Chemistry at Fuzhou University, Fuzhou, Fujian 350108, China. [2]Department of Chemistry and Shenzhen Grubbs Institute, Guangdong Provincial Key Laboratory of Catalysis, Southern University of Science and Technology, Shenzhen, Guangdong 518055, China. [3]School of Chemistry and Chemical Engineering, Henan Normal University, Xinxiang, Henan 453007, China. [4]These authors contributed equally: Kai Yang, Yanfei Mao, Zhihan Zhang. ✉e-mail: yupy@sustech.edu.cn; qsong@fzu.edu.cn

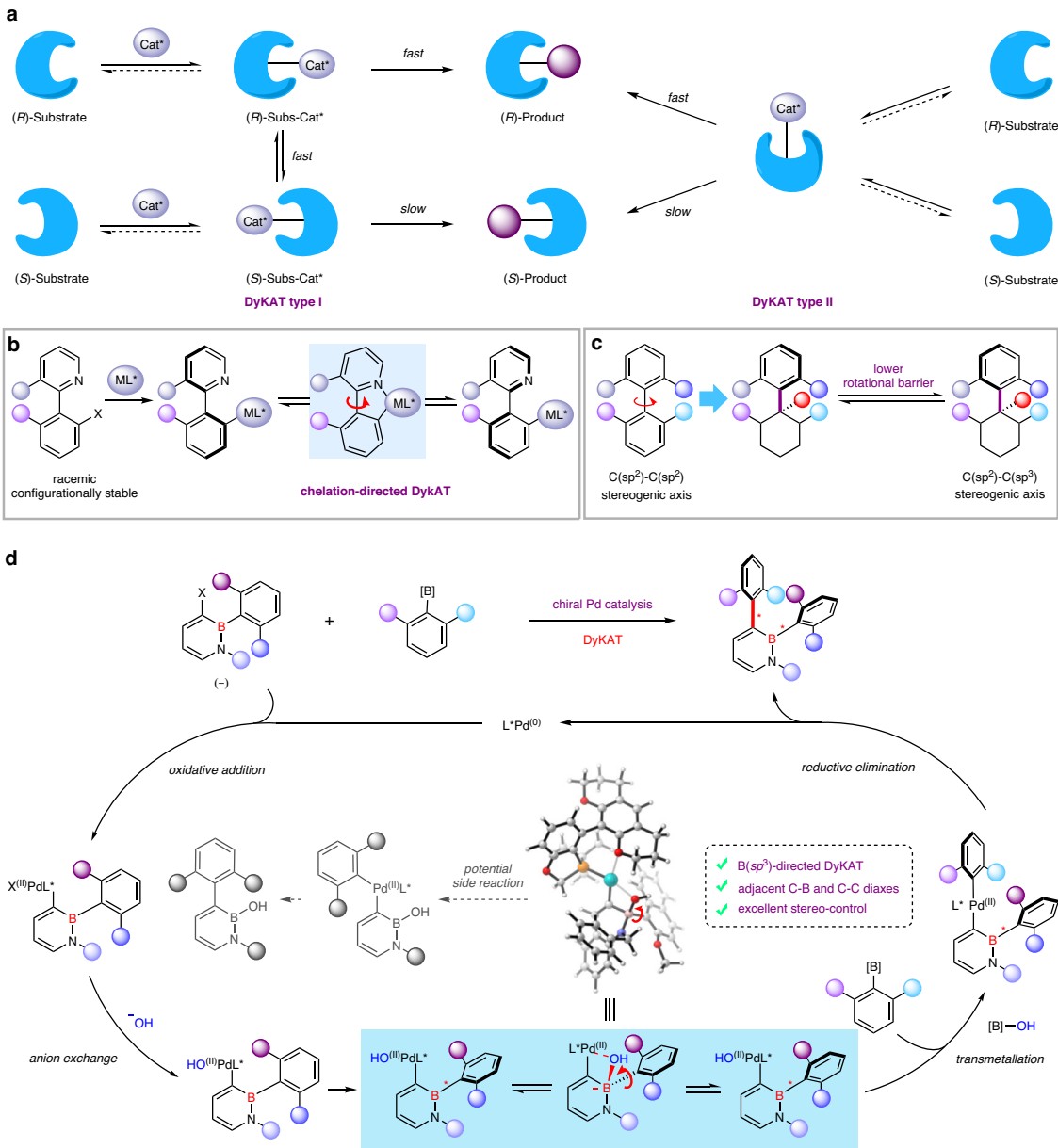

**Fig. 1 | Previous transition-metal-catalyzed atroposelective cross-coupling reactions and our reaction design. a** Dynamic kinetic asymmetric transformation (DyKAT). **b** Chelation-directed DyKAT of racemic heterobiaryls. **c** C(sp²)−C(sp³) stereogenic axis. **d** A tetracoordinate boron-directed DyKAT to access the challenging atropisomers bearing C-B stereogenic axis, or adjacent C-B and C-C diaxes (this work).

reversible, it might be a new model for the interconversion of dia-stereomeric intermediates in DyKAT. As a result of our continuous interest in tetracoordinate boron chemistry[34], we envisioned that the reversibility between B(sp²) and B(sp³)[35–40] could support B(sp³)-directed DyKAT and fabricate optically pure C-B axially chiral molecules[41,42] (Fig. 1d), which as elusive atropisomers and this type of chiral orga-noborons are underdeveloped and represent a big hurdle and chal-lenge in boron chemistry as well as in axial chirality compared to their congeners with C-C or C-N axis, owing to the lower rotational barrier which is caused by longer C-B bond[43–47]. If successful, this reaction would develop an interesting diastereomeric intermediate equilibrium process that differs from previous chelation-directed DyKAT of racemic heterobiaryls[18–25]. Herein, we present a palladium-catalyzed dynamic kinetic asymmetric cross-coupling of racemic, configurationally stable 3-bromo-2,1-azaborines[48–53] with boronic acid derivatives via an equilibrium mode of DyKAT mediated by

tetracoordinate boron intermediates. By doing so, the DyKAT strategy could be employed to the assembly of challenging atropisomers with C-B axis or adjacent C-B and C-C diaxes.

On the basis of the previous reports of DyKAT[2,7] and asymmetric Suzuki–Miyaura coupling reactions[54–67], our proposed catalytic DyKAT version is shown in Fig. 1d. Oxidative addition of racemic 3-bromo-2,1-azaborines to chiral Pd(0) species and subsequent anion exchange afford diastereomeric intermediate **I**. The fast equilibration of inter-mediate (**R**)-**I** and (**S**)-**I** could occur through the tetracoordinate boron intermediate **II** formed by the transfer of the hydroxy group from Pd[68]. Then, the transmetalation between intermediate **I** and boronic acid derivatives and final reductive elimination generates C-B axial chirality and regenerates the chiral Pd(0) catalysis. It is important that one of the diastereomeric intermediate **I** ((**R**)-**I** or (**S**)-**I**) undergoes the trans-metalation step faster than the other, so as to achieve the goal of DyKAT. Although mechanistically appealing, there are several

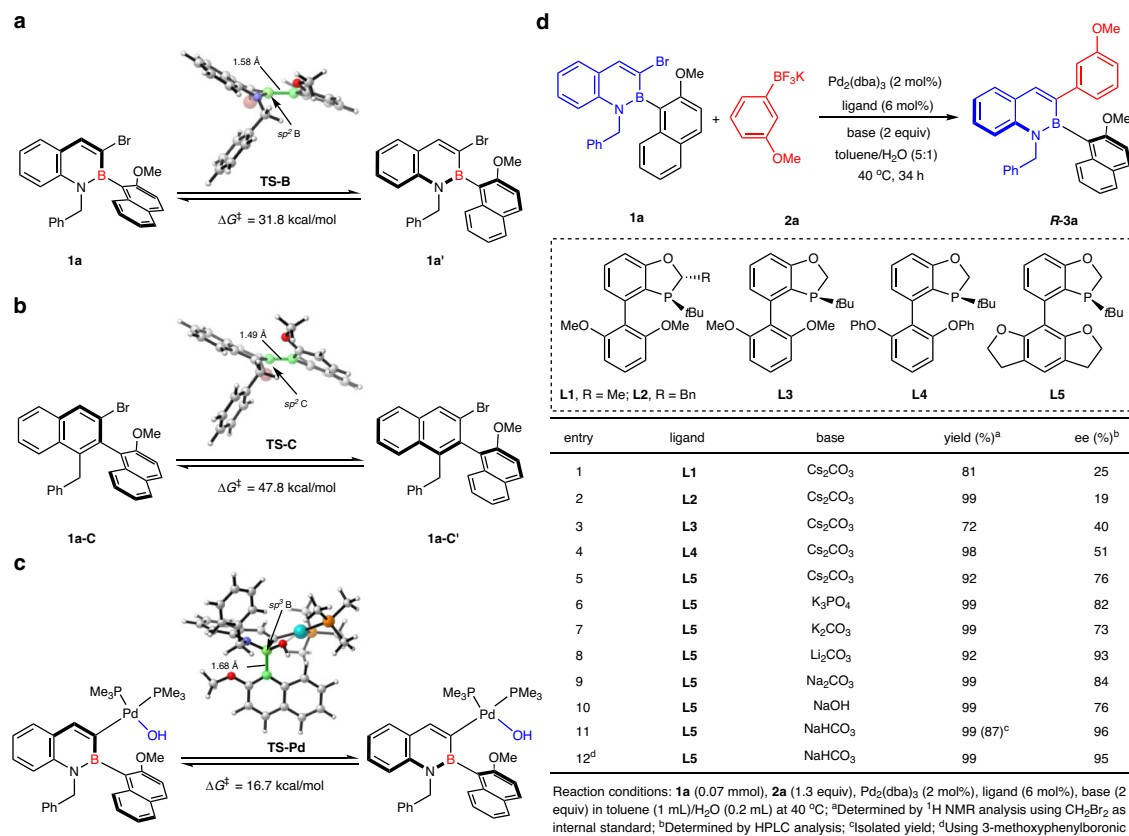

**Fig. 2 | Preliminary density functional theory (DFT) calculations on the racemization processes and condition optimization. a** Activation energy barrier for axial rotation of 3-bromo-2,1-borazaronaphthalene **1a. b** Activation energy barrier for axial rotation of 3-bromo-naphthalene **1a-C. c** Activation energy barrier for axial rotation of 3-Pd-2,1-borazaronaphthalene via a TS featuring tetracoordinate boron geometry, **TS-Pd. d** Condition optimization for palladium-catalyzed dynamic kinetic asymmetric cross-coupling.

considerable challenges: (1) the sterically hindered environment around B atom may inhibit the formation of tetracoordinate boron intermediates; (2) it is still uncertain whether the tetracoordinate boron intermediate could really reduce the rotation barrier and facilitate rotation of the aryl group on B atom around the C-B stereogenic axis; (3) competitive intramolecular self-coupling side reactions might occur[69]; (4) the simultaneous diastereoselective and enantioselective synthesis of axially chiral molecules with multiple axes by one-step reactions is still in its infancy[70–74].

## Results

To validate our hypothesis, we first designed and synthesized racemic 3-bromo-2,1-borazaronaphthalene **1a**. Preliminary density functional theory (DFT) calculations were performed to evaluate the feasibilities of the racemization processes of three different species. As depicted in Fig. 2a, substrate **1a** with a C−B axis has a rotation barrier of 31.8 kcal/mol to **1a'** since the congested steric environment of the planar geometry in transition states induces large distortions of aromatic rings. Compared with **1a**, substrate **1a-C** for the traditional asymmetric Suzuki-Miyaura coupling possesses not only a shorter C−C axis but also stronger aromaticity, which renders a much higher rotational barrier of 47.8 kcal/mol to **1a-C'**, making direct dynamic kinetic asymmetric transformation from the substrate (DyKAT) even more unattainable (Fig. 2b). However, the Lewis acidic boronic complex **1a-Pd**, the intermediate after oxidative addition of **1a** to Pd followed by ligand exchange, allows the coordination of the hydroxide ligand to form a chiral tetracoordinate boron species. In **TS-Pd**, the tetracoordinate boron species own elongated C−B axis. The corresponding rotational barrier from **1a-Pd** to **1a-Pd'** is significantly reduced to

16.7 kcal/mol (Fig. 2c), which makes the free rotation of the aryl group around C−B stereogenic axis feasible and fully supports our conjecture.

Encouraged by the results from theoretical calculations, we then investigated this envisioned dynamic kinetic cross-coupling using racemic 3-bromo-2,1-borazaronaphthalene **1a** and trifluoroborate **2a** as model substrates (Fig. 2d). Delightfully, this reaction with Pd₂(dba)₃ as catalyst, the P-chiral monophosphorus ligand **L1** as ligand and Cs₂CO₃ as base in toluene/H₂O furnished the desired C-B axially chiral product (**R**)-**3a** in 81% NMR yield with 25% enantioselectivity excess (ee) at 40 °C (Fig. 2d, entry 1). This result proved the feasibility of our hypothesis and encouraged us to further evaluate other ligands. The P-chiral monophosphorus ligands with small steric hindrance led to higher ee values (Fig. 2d, entries 1–3). The substituents on the aryl units of the ligands have an effect on this reaction (Fig. 2d, entries 4-5), and a better result (Fig. 2d, entry 5, 92% yield and 76% ee) was obtained when ligand **L5** with tetrahydrobenzofuran group was used. Subsequently, we investigated the effect of bases and found that these bases all promoted this reaction well, but the enantioselectivities of this reaction were sensitive to bases (Fig. 2d, entries 6–10). In general, weak bases were more favorable for enantioselectivities than strong bases. Overall, the optimized reaction conditions for this DyKAT are shown below: **1a** (1 equiv), **2a** (1.3 equiv), Pd₂(dba)₃ (2 mol%), **L5** (6 mol%), NaHCO₃ (2 equiv) in toluene/H₂O at 40 °C for 34 h (Fig. 2d, entry 11). In addition, the same yield and enantioselectivity were obtained by reducing the proportion of water when 3-methoxyphenylboronic acid (**2a'**) was used as the substrate (Fig. 2d, entry 12).

To better understand the racemization process of the DyKAT, the following experiments were performed. As illustrated in Fig. 3a, the profile of the ee values or yields of the recovered **1a** and the product **3a**

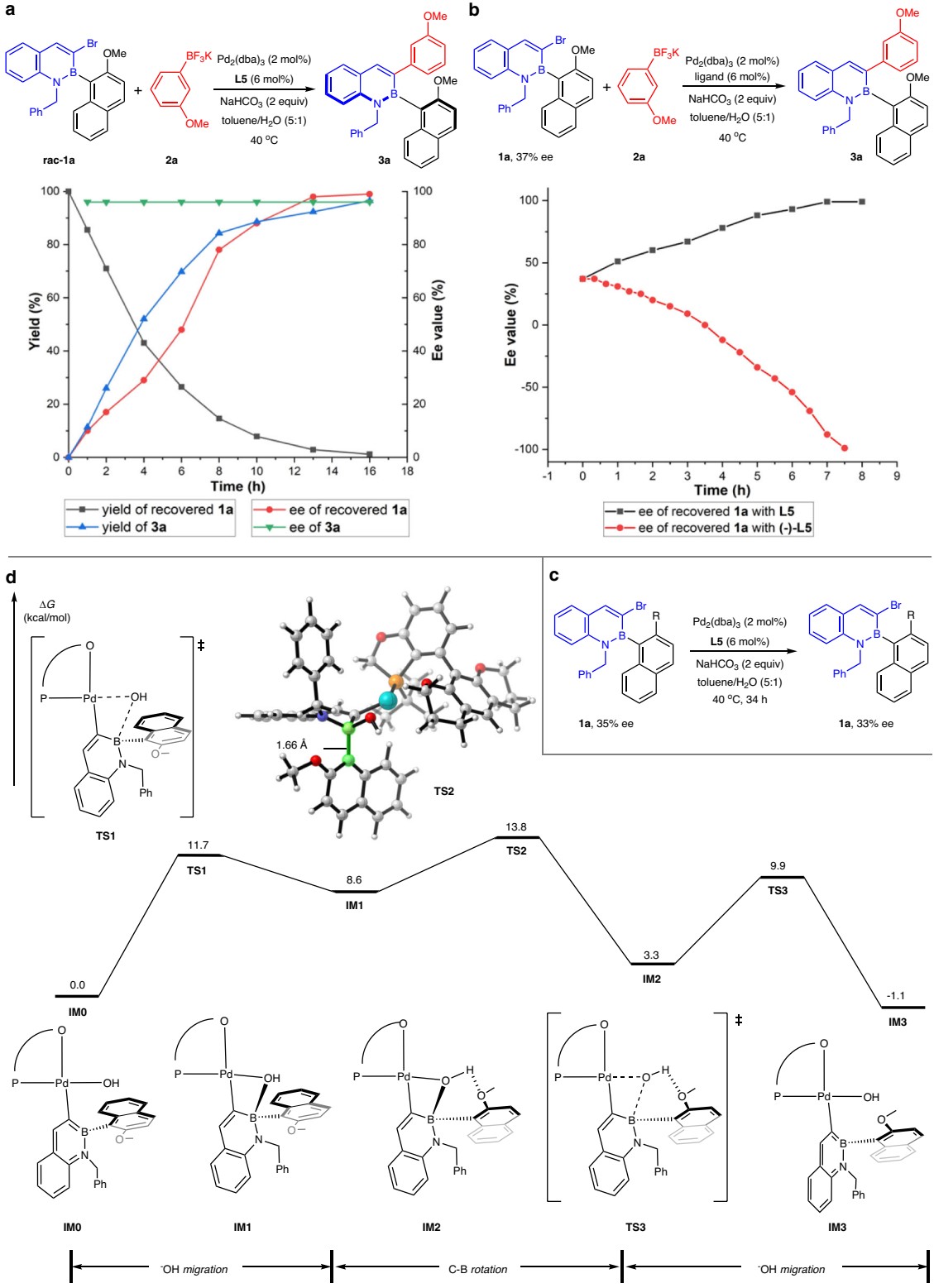

**Fig. 3 | Mechanistic studies. a** Yields and ee values of product **3a** (recovered **1a**) at different reaction times. **b** Ee values of recovered **1a** at different reaction times by using ligands **L5** and **(-)-L5**. **c** Racemization experiment under standard conditions without aryl trifluoroborates. **d** Free energy profile calculated for the racemization process with **L5** as ligand.

versus time indicated that two enantiomers of **1a** were consumed together and one of the enantiomers was decreased more rapidly, suggesting a kinetic resolution (KR) process. In addition, the reactions of enantioenriched **1a** (37% ee) with two ligands with different configurations were carried out, and the profile of the ee values of the recovered **1a** versus time was shown in Fig. 3b. The results also supported a KR process. Finally, no obvious racemization of enantioenriched 3-bromo-2,1-azaborine **1a** under standard conditions without aryl trifluoroborates, excluding a dynamic kinetic resolution (DKR) pathway. To demonstrate that the process is indeed a DyKAT, DFT calculations

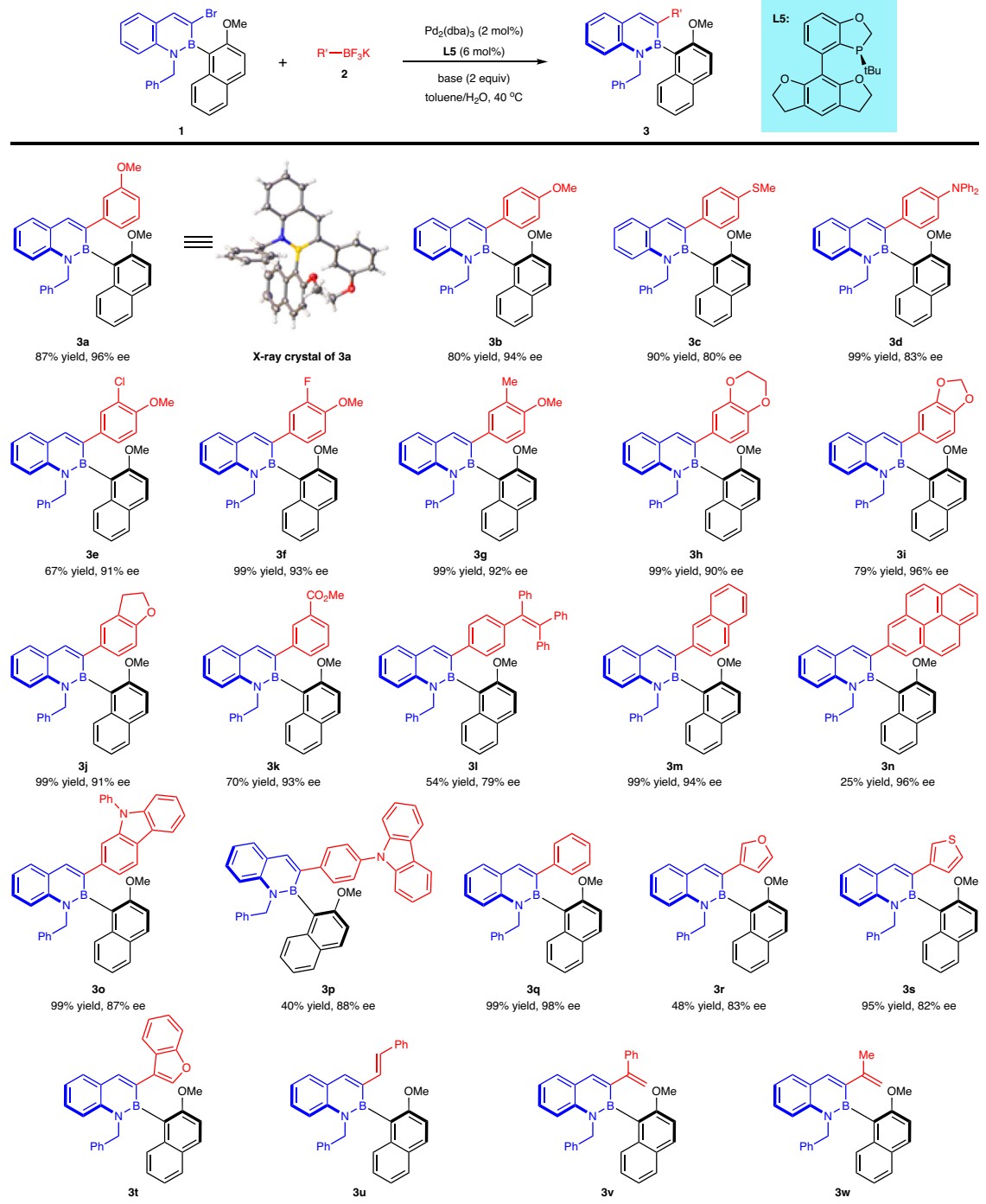

**Fig. 4 | Substrate scope for trifluoroborates.** Reaction conditions: **1** (0.1 mmol), **2** (0.13 mmol), Pd₂(dba)₃ (2 mol%), **L5** (6 mol%), NaHCO₃ (2 equiv) in toluene (1.5 mL)/H₂O (0.3 mL) at 40 °C; isolated yields are provided.

were performed to probe the mechanism of the racemization process. After oxidative addition and anion ligand exchange, benefiting from the boron Lewis acidity, **IM0** first underwent an intramolecular hydroxide migration to form a tetracoordinate boron species **IM1** via **TS1**. The C-B bond in **IM1** is free to rotate with a small barrier of 5.2 kcal/mol. The analysis of the geometry of **TS2** indicates that owing to the formation of tetracoordinate structure, the naphthalene moiety undergoing rotation is placed on the axial position to avoid repulsions with the benzylic group sprouted on the equatorial position. Meanwhile, the C-B bond is elongated by ~0.1 Å, which also provides more space to relax the strain in **TS2**. Interestingly, **IM2** is more stable than its diastereomer **IM1** due

to the formation of an intramolecular hydrogen bond. The overall energy barrier for the racemization process is 13.8 kcal/mol, endorsing our strategy that the rotation around C–B axis could be realized even with very bulky ligands.

Applying the optimized reaction conditions to a range of substrates demonstrates the generality of this DyKAT (Fig. 4). This approach was compatible with aryl trifluoroborate bearing electron-rich groups, including alkoxy (**3a**, **3b**, and **3e**–**3j**), methylthio (**3c**), and *N,N*-diphenyl (**3d**), delivering the corresponding C-B axially chiral products in high yields with good to excellent enantioselectivities (80–96% ee). The absolute configuration of (*R*ₐ)-**3a** was determined by

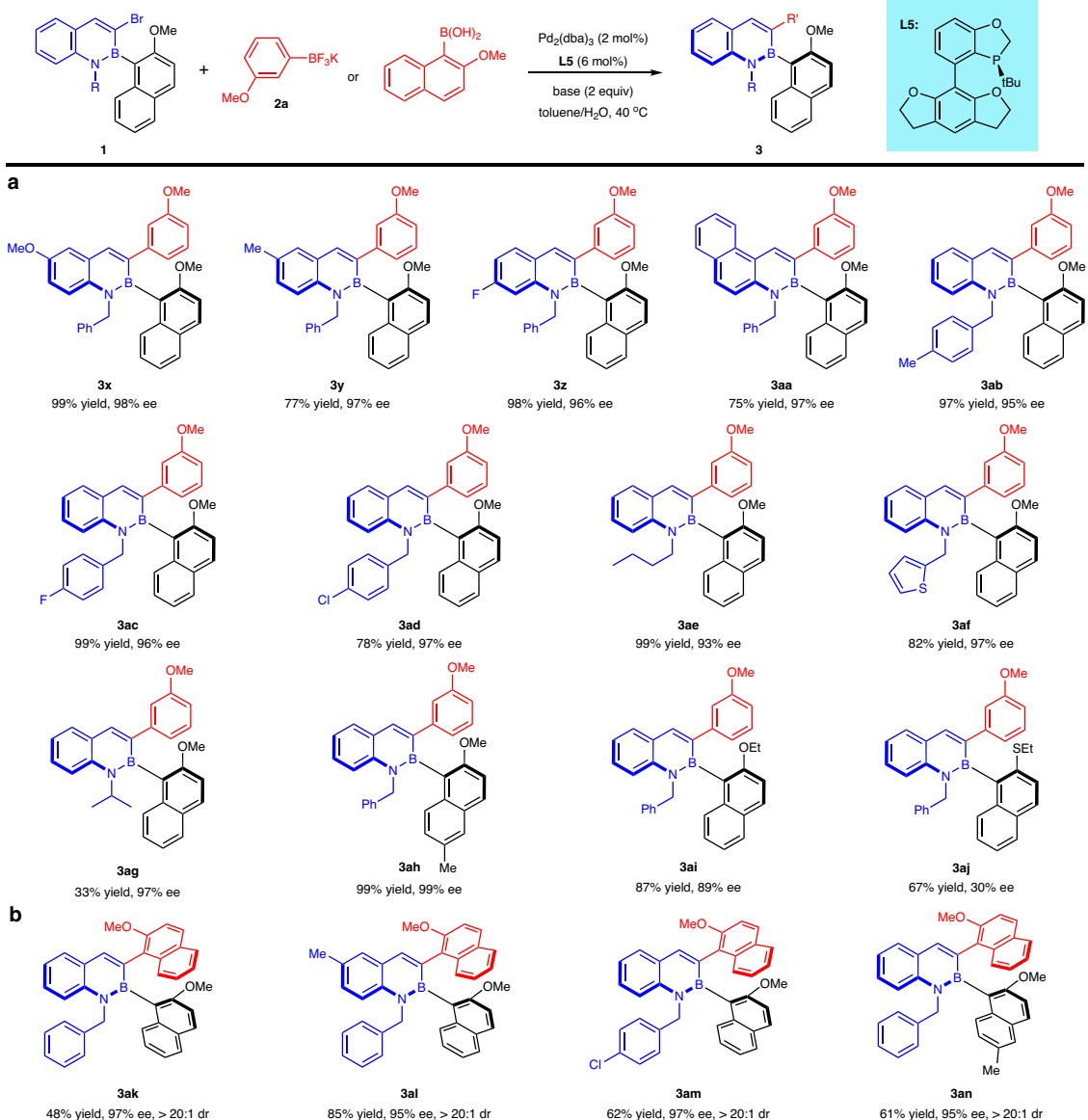

**Fig. 5 | Substrate scope for racemic 3-bromo-2,1-borazaronaphthalenes and diaxially chiral compounds. a** Scope for racemic 3-bromo-2,1-borazar-onaphthalenes. Reaction conditions: **1** (0.1 mmol), **2a** (0.13 mmol), Pd₂(dba)₃ (2 mol%), **L5** (6 mol%), NaHCO₃ (2 equiv) in toluene (1.5 mL)/H₂O (0.3 mL) at 40 °C; isolated yields are provided. **b** Scope of adjacent C-B and C-C diaxially chiral compounds. Reaction conditions: **1** (0.1 mmol), 2-methoxy-1-naphthyl) boronic acid (1.3–4.0 equiv), Pd₂(dba)₃ (2 mol%), **L5** (6 mol%), Li₂CO₃ (2.0–4.0 equiv) in toluene (1.5 mL)/H₂O (0.15 mL) at 40 °C; isolated yields are provided.

X-ray crystallographic analysis (CCDC 2245394, the CIF file is provided in Supplementary Data 1). Aryl trifluoroborate with an electron-withdrawing group was tolerated well under the standard conditions (**3k**, 70% yield and 93% ee). The tetrastyryl group could also be introduced into the desired product **3l** by this method, which provides the possibility for a chiral AIE molecule. Polycyclic aryl trifluoroborates (**3m** and **3n**) and unsubstituted phenyl trifluoroborate (**3q**) were successfully coupled with excellent enantioselectivities to desired products. Moreover, aryl trifluoroborates bearing heteroaromatic components, including carbazoles (**3o** and **3p**), furan (**3r**), thiophene (**3s**), and benzothiophene (**3t**), could be smoothly converted into the target products with good to excellent enantioselectivities (82–96% ee). Alkenyl trifluoroborates underwent this reaction well, and the better enantioselectivities of 1-substituted alkenyl trifluoroborates (**3v** and **3w**) than (*E*)-styryl trifluoroborate (**3u**) may be due to steric hindrance.

Next, a wide range of racemic 3-bromo-2,1-borazaronaphthalenes could all undergo this DyKAT to render the corresponding enantiomerically enriched C-B axially chiral molecules (Fig. 5a). Methoxy (**3x**), methyl (**3y** and **3ah**), and fluoro (**3z**)-substituted racemic 3-bromo-2,1-borazaronaphthalenes could successfully deliver the desired products in excellent efficiency (77–98% yields and 96–98% ee). Notably, BN-phenanthrene (**3aa**) was a viable framework for this transformation, providing the corresponding product with excellent enantioselectivity. Moreover, substituents on the N atom of the 2,1-borazaronaphthalene including benzyls (**3ab-3ad**), *n*-butyl (**3ae**), and thiophen-2-ylmethyl (**3af**) were readily tolerated well. Despite lower yield, the transformation also tolerated bulky (iso-propyl) moiety on the N atom of the 2,1-borazaronaphthalene with excellent enantioselectivity (**3ag**, 33% yield and 97% ee). Low enantioselectivities were obtained when the OMe group was changed to OEt (**3ai**) or SEt (**3aj**) groups with larger steric hindrance.

In view of the successful application of the DyKAT strategy to prepare the C-B axially chiral compounds, we turned our attention to the synthesis of atropisomers with C-B adjacent diaxes of C-B and C-C bonds (Fig. 5b). (2-Methoxy-1-naphthyl)boronic acid was tested in the

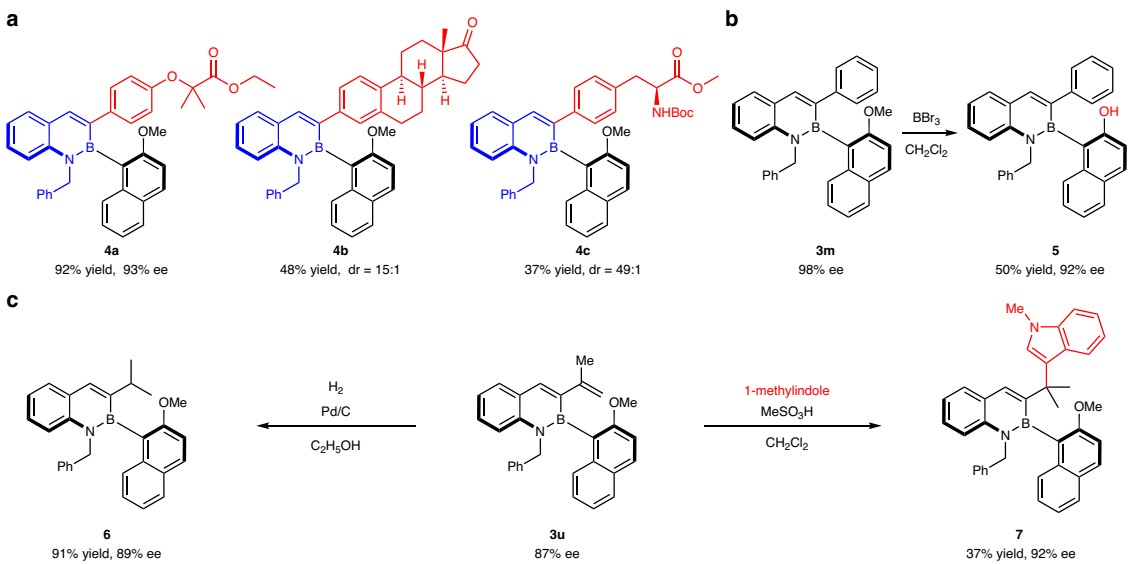

**Fig. 6 | Functionalization of complex molecules and synthetic transformations. a** Functionalization of complex molecules. **b** Demethylation of **3m**. **c** Palladium-catalyzed hydrogenation of **3u** and Brønsted acid-catalyzed alkylation of indole with **3u** as alkylation reagent. Boc = *t*-butoxycarbonyl.

reaction, and to our delight, the desired axially chiral products **3ak**-**3an** were obtained with excellent diastereoselectivities and enantioselectivities (>20:1 dr, 95–97% ee). The absolute configuration of **3ak** was determined by ECD and two-dimensional NMR experiments (for details, see Supplementary Figs. 1–4 and 6–10)[75–77].

This transformation is also applicable to the synthesis of C-B axially chiral compounds bearing complex fragments derived from natural products or therapeutic agents, whose high functional-group compatibility is fully linchpinned. Aryl trifluoroborates derived from clofibrate (**4a**), estrone (**4b**), and tyrosine (**4c**) were transformed into the corresponding C-B axially chiral compounds with ease (Fig. 6a). In addition, C-B axially chiral compounds could be further modified. Firstly, demethylation of product **3m** could generate a C-B axially chiral molecule **5** with free naphthol, which has the potential for further transformations (Fig. 6b). Meanwhile, product **3u** could be converted to isopropyl-substituted C-B axially chiral molecule **6** via hydrogenation, and could also react with indole under acid catalysis to afford compound **7** with high retention of the enantiopurity (Fig. 6c).

In conclusion, we developed a palladium-catalyzed DyKAT process of racemic, configurationally stable 3-bromo-2,1-azaborines for the construction of C-B axial chirality. The experiments and calculations demonstrated that the reaction is a DyKAT process and that the reversible tetracoordinate boron intermediate is the key to its success. This chemistry offers practical access to chiral organoborons bearing C-B axis or adjacent C-B and C-C diaxes in generally high yields with excellent diastereoselevtivities and enantioselctivities.

## Methods

### General procedure for the synthesis of atropisomers with a single C-B stereogenic axis

In air, a 25 mL Schlenk tube was charged with **1** (0.1 mmol, 1 equiv), **2** (0.13 mmol, 1.3 equiv), Pd$_2$(dba)$_3$ (2 mol%), **L5** (6 mol%), and NaHCO$_3$ (0.2 mmol, 2.0 equiv). The tube was evacuated and filled with argon for three cycles. Then, 1.5 mL of toluene and 0.3 ml of water was added under argon. The reaction was allowed to stir at 40 °C for 34 h. Upon completion, a proper amount of silica gel was added to the reaction mixture. After the removal of the solvent, the crude reaction mixture was purified on silica gel (petroleum ether and ethyl acetate) to afford the desired products.

### General procedure for the synthesis of atropisomers with adjacent diaxes of C-B bond and C-C bond

In air, a 25 mL Schlenk tube was charged with **1** (0.1 mmol, 1 equiv), 2-methoxy-1-naphthyl)boronic acid (1.3–4.0 equiv), Pd$_2$(dba)$_3$ (2 mol%), **L5** (6 mol%), and Li$_2$CO$_3$ (2.0–4.0 equiv). The tube was evacuated and filled with argon for three cycles. Then, 1.5 mL of toluene and 0.15 ml of water was added under argon. The reaction was allowed to stir at 40 °C for 46–76 h. Upon completion, a proper amount of silica gel was added to the reaction mixture. After the removal of the solvent, the crude reaction mixture was purified on silica gel (petroleum ether and ethyl acetate) to afford the desired products.

## Data availability

The data that support the findings of this study are available within the article and its Supplementary Information files. All other data are available from the corresponding author upon request. Supplementary Tables 1 and 2 for mechanism experiment results, Supplementary Table 3 for rotational barrier of **3a**, Supplementary Figs. 1–4 and 10 for additional computational results, Supplementary Fig. 5 for the plot of ln(ee$_0$/ee$_t$) vs time of **3a**, Supplementary Figs. 6–9 for two-dimensional NMR analysis of **3ai**, Supplementary Figs. 11–237 for NMR spectra, Supplementary Figs. 238–283 for HPLC spectra. The X-ray crystallographic coordinates for the structure reported in this study have been deposited at the Cambridge Crystallographic Data Centre (CCDC), under deposition number 2245394 (**3a**). These data can be obtained free of charge from The Cambridge Crystallographic Data Centre via www.ccdc.cam.ac.uk/data_request/cif. The cartesian coordinates of the optimized structures are provided in a source data file. Source data are provided with this paper.

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

## Acknowledgements

Financial support from National Natural Science Foundation of China (21931013, 22001038, 22271105, and 22271048), the Natural Science Foundation of Fujian Province (2022J02009 and 2022J05016), Fuzhou University (510578), Open Research Fund of School of Chemistry and Chemical Engineering (Henan Normal University), Guangdong Provincial Key Laboratory of Catalysis (2020B121201002), and Shenzhen Higher Education Institution Stable Support Plan (20200925152921001) is gratefully acknowl-edged. Computational work was supported by the Center for Computational Science and Engineering at the Southern University of Science and Technology and the CHEM high-performance supercomputer cluster (CHEM-HPC) located at the Department of Chemistry, Southern University of Science and Technology.

## Author contributions

Q.S. conceived and directed the project. K.Y. and Y.M. performed experiments and prepared supplementary information. J.X., H.W., and Y.H. helped collect some new compounds and analyze the data. P.Y. and Z.Z. performed the DFT calculations and drafted the DFT parts. Q.S., P.Y., K.Y., and Z.Z. wrote the paper. All authors discussed the results and commented on the manuscript.

## Competing interests

The authors declare no competing interests.
