## [Peer Review File · Nature Communications]

Construction of C-B axial chirality via dynamic kinetic asymmetric cross-coupling mediated by tetracoordinate boronReviewers' Comments:

Reviewer #1:

Revision.

This manuscript describes the interesting DyKAT to obtain atropisomeric C-B axial chirality.

DyKAT has been known to controlling Csp²-Csp² atropisomeric resolution, but to my knowledge it is the first time that is applied to C-B bond. An intermediate tetracoordinate boron was generated in situ, significantly lowering the atropisomerization barrier. The racemization and the subsequent enantioselective coupling makes it possible to carry out the reaction by selecting a single atropisomer.

Although, this strategy is of interest, I recommend major revision.

Specifically, the nomenclature descriptors (P and M) of the atropisomers was not assigned. I did not understand which dihedrals for the two chiral axes were taken into consideration to calculate the ECD. TD-DFT calculations of ECD should be done with different functional for data reproducibility and more NStates (i.e. nstates=70) to also consider the naphthyl absorption region around 220-240 nm. For the same reason, the experimental ECD spectrum could be done in acetonitrile to record the region between 190-400 nm.

The assignment of absolute configuration of a compound with a single atropisomeric axis should be done. The X-ray structure of 3a was acquired with Mo-K α radiation, which is unreliable when the molecules lacking sufficiently heavy atoms ($Z > Si$). Indeed, the Flack parameter has an uncertainty of 0.7, which means the results are meaningless. TD-DFT calculations should be used also in this case.

The conformational research of ground states and transition states compounds has not properly been done or explained in detail. Coordinates for TSs and negative frequencies are not reported.

The HPLC should be revised. In particular: Why has rac-3q two enantiomers and four peaks? Why have Rac-3ai, 3aj, 3ak, 3al two diastereoisomers and only two peaks? Moreover, it is apparently strange that a diastereomerization (i.g. Rac-3ai) yields 50% of each diastereoisomer. Syn/Anti

disposition of Methoxy group could give a significantly different ratio. DFT calculations might help explain it.

Please also consider the points below:

Introduction

- 1) Please Remove “And” at the beginning of the sentence “And transition-metal-catalyzed Dykat”

Results

- 2) Regarding the sentence:

“However, the Lewis acidic boronic complex 1a-Pd, the intermediate after oxidative addition of 1a to Pd followed by ligand exchange, allows the coordination of the hydroxide ligand to form a tetracoordinate boron species with an elongated C–B axis. The corresponding rotational barrier from 1a-Pd to 1a-Pd’ is significantly reduced to 16.7 kcal/mol (Fig. 2c), which makes the free rotation of the aryl group around C–B stereogenic axis feasible, and fully support our conjecture.”

I didn’t find how they have calculated GSs of 1a-Pd, 1a-Pd’ and the relative TS. Please introduce in the text or in the SI a better discussion of it.

- 3) In Figure 2 the chemdraw structure of 1a-Pd not shows the coordination between Boron and OH.
- 4) Please change “Finally, reactions with **chiral** 3-bromo-2,1-azaborine 1a were carried out,” with “Finally, reactions with **enantioenriched** 3-bromo-2,1-1a were carried out,”
- 5) Please Change “(3a and (-)-3a)” with “(3a and **M**-3a)”
- 6) To reinforce their conjecture, they could do a kinetic experiment starting with 3a enantiopure with the two different ligands.
- 7) In Figure 4, the compounds 3ai, 3aj, 3ak, 3al are diastereoisomers, they should insert the d.r.

Supporting Information

- 1) ^{19}F NMR is missing. The CH_2 of Bn is an AB system in chiral environment. That mean they are two different diastereotopic protons. Sometime it was written as quartet, instead of two distinct doublets.
- 2) The ^{11}B NMR spectra could be better done in a quartz tube to remove or reduce the broaden band at 0 due to the borosilicates of glass and/or of probe material.

Reviewer #2:

Remarks to the Author:

The manuscript from Song, Yu et al. entitled "Construction of C-B axial chirality via dynamic kinetic asymmetric cross-coupling mediated by tetracoordinate boron" describes the enantioselective synthesis of azaborines bearing a C-B chiral axis, using a Pd-catalyzed dynamic kinetic asymmetric transformation (DyKAT). The methodology was extended to the concomitant formation of both a C-B and C-C chiral axis. The corresponding cyclic organoborons are usually obtained with excellent yields and enantioselectivities. According to the authors, and supported by DFT calculation, the formation of a tetracoordinate boron intermediate is the key of the success of this transformation.

The preparation of atropisomers bearing a C-B chiral axis is a very challenging task, with only few examples described (to my knowledge 6 including 3 asymmetric synthesis). In the context of C-B atropisomers synthesis, the authors present an original approach based on DyKAT, a strategy that has been recently described by Stoltz and Lasseletta for the atroposelective synthesis of heterobiaryls. This is very interesting to apply this concept to the formation of C-B chiral axis, and it is a significance work in the field of atropisomers.

I have however, some concerns regarding the way the authors present this work. Indeed, the whole hypothesis relies on the formation of a tetracoordinated boron atom that apparently allows the DyKAT on these particular substrates. This is only supported by DFT calculations. I am not fully convinced by this hypothesis, and I think additional experimental data are required to confirm or not this hypothesis.

First, only substrates with a 2-methoxy group on the naphthyl moiety are used. DFT calculations suggest that the OMe is involved in the DyKAT. What about other substituents? Some substrates with a non-coordinating group should be tested to strengthen the calculations. One can argue that a coordination could also occur between the palladium and the OMe group. Has this hypothesis been rejected? It would be interesting to calculate the rotational barrier involving such coordination. What happen when a bulkier OR group is used, that could prevent the coordination?

For the scope of the reaction, other groups that methoxy should be evaluated in the transformation (ie. OEt, OPh, SMe, ...).

In figure 3a, the yields and ee of product 3a, and recovered SM 1a are plotted versus time. The authors should indicate in the text the initial ee for this experiment (racemic ?). Since they use SM 1a with 33%ee in figure 3b and c, it could be a bit confusing.

In figure 3a, the authors suggest a kinetic resolution and then a dynamic kinetic resolution. How do they draw this conclusion? When does the dynamic kinetic resolution started on Fig 3a ? At 12h hours, the ee of 1a is about 63%ee, then it drops down to ~33%ee, which means that 1a is racemized. First, I don't understand why the ee of SM1a is about 33% at the end of the reaction. Second, how 1a is racemized ? The authors shows in Fig 3b that SM 1a is not racemized with NaHCO₃ in toluene/water. Does it racemize under standard condition (with NaHCO₃, Pd/ligand, Tol/H₂O) without the aryIBF3K ? Does the oxidative addition of the palladium reversible? (see J. Organomet. Chem. 2020, 121571). Additional experiments should be done, and more explanations provided to help the reader understand.

Otherwise, the experimental part is well written with all the requested data. However, I didn't find the atom coordinates for IM1. Appropriate references are also included.

Other comments:

- 1) For the synthesis of compounds with 2 chiral axes, Li₂CO₃ was used, but this base hasn't been tested in Fig2d. What is the results with Li₂CO₃ on the model substrate?
- 2) Fig 3c. What is the conclusion of this experiment?
- 3) The absolute configuration was determined by X-ray of 3a. Is it consistent with the DFT calculations?
- 4) Fig 4 : compounds 3n, 3p, 3r, 3ag, 3ai with yields below 50%. Since yields are lower than 50%, is it a DyKAT process or a kinetic resolution?

Reviewer #3:

Remarks to the Author:

In this manuscript, Song, Yu and co-workers report a novel dynamic kinetic asymmetric transformation (DyKAT), accomplished via a tetracoordinate boron-directed diastereomeric intermediate equilibrium process that differs from previous chelation-directed DyKAT of racemic heterobiaryls. This equilibrium process is well designed, innovative and effective, and it is also supported by DFT calculations and control experiments. Through this methodology, a variety of C-B axially chiral compounds were prepared in high enantioselectivity. More interestingly, atropisomers with C-B and C-C diaxes were also prepared in high diastereoselectivity and enantioselectivity. This methodology has also been applied to complex molecules. The manuscript and supplementary information were well written and prepared. For these reasons, I believe that this work is suitable for publication in Nature Communications after consideration of the following comments:

1. DFT calculations well supported the design of this interesting dynamic kinetic asymmetric transformations and helped determine the absolute configuration for adjacent C-B and C-C diaxially chiral compound. The energy barrier of rotation around the C-B bond in 1a-Pd was calculated to be much lower than that in the substrate, suggesting the transformation between two different configurations is enabled by the Pd catalyst. This rotation relies on the coordination of hydroxide ligand. I was wondering whether the ligand exchange from bromide to hydroxide is thermodynamically feasible. Considering the significant role of the hydroxide, I suggest that the authors should try their best to search for more evidence and details from the literature or perform further calculations to support the possibility of the formation of Pd-OH complex.
2. In abstract, the authors should use the more specific "diastereo- and enantio-control" instead of "stereo-control".
3. Fig. 2d caption, the authors should add NMR Internal standard.
4. Page 5, line 18, "Pd2(dba)2" should be replaced by "Pd2(dba)3".
5. Page 7, line 1, "stereoselectivities" should be replaced by "enantioselectivity".
6. Why is the yield of product 3ag so low? Is kinetic resolution involved here?
7. How about using alkyl trifluoroborates as the substrate? Some commentary should be added.
8. In supplementary information, the authors should add dr values of compounds 4b and 4c.

Point-by-point Response to Reviewer(s)' Comments

Reviewer #1

This manuscript describes the interesting DyKAT to obtain atropisomeric C-B axial chirality. DyKAT has been known to controlling Csp²-Csp² atropisomeric resolution, but to my knowledge it is the first time that is applied to C-B bond. An intermediate tetracoordinate boron was generated in situ, significantly lowering the atropisomerization barrier. The racemization and the subsequent enantioselective coupling makes it possible to carry out the reaction by selecting a single atropisomer. Although, this strategy is of interest, I recommend major revision.

Response: We deeply thank this reviewer for the favorable comments on our work, we really appreciate it.

Specifically, the nomenclature descriptors (P and M) of the atropisomers was not assigned.

Response: We thank this reviewer for pointing it out. Since R/S has been widely used to label atropisomers in axial chirality in literature (for example: *Nat. Chem.* **9**, 558–562 (2017); *Nat. Catal.* **3**, 727–733 (2020); *Nat. Catal.* **2**, 504–513 (2019)), we also used R/S to label our atropisomers in the manuscript as well as in the supporting information for consistency. Please see our revised materials.

I did not understand which dihedrals for the two chiral axes were taken into consideration to calculate the ECD.

Response: We express our gratitude to the reviewer for bringing this to our attention. To make it clearer, we have included a figure highlighting the corresponding chiral axes in both the revised manuscript and Supporting Information. Please see our revised manuscript and Supporting Information.

TD-DFT calculations of ECD should be done with different functional for data

reproducibility and more NStates (i.e. nstates=70) to also consider the naphthyl absorption region around 220-240 nm. For the same reason, the experimental ECD spectrum could be done in acetonitrile to record the region between 190-400 nm.

Response: We extend our thanks to the reviewer for the suggestions. In response, we have incorporated CAM-B3LYP, which is also a suitable functional for TD-DFT calculations. Additionally, we have plotted a spectrum based on data obtained using CAM-B3LYP, and although it exhibits a similar trend with wB97XD, the latter has a closer resemblance to experimental results (see Figure below).

We have further conducted TD-DFT calculations, this time incorporating more states (Nstates = 70) for the most favorable conformation, and found that plotted spectrums using different parameters were approximately equal to those calculated using nstates = 30, denoting that using nstates = 30 should be accurate for our system.

The assignment of absolute configuration of a compound with a single atropisomeric axis should be done. The X-ray structure of 3a was acquired with Mo-K α radiation, which is unreliable when the molecules lacking sufficiently heavy atoms ($Z > \text{Si}$).

Indeed, the Flack parameter has an uncertainty of 0.7, which means the results are meaningless.

Response: We thank this reviewer for pointing this out. The data of X-ray structure of **3a** has been collected by using Mo-K α radiation, and the new Flack parameter is 0.06. Please see our revised Supporting Information.

TD-DFT calculations should be used also in this case. The conformational research of ground states and transition states compounds has not properly been done or explained in detail. Coordinates for TSs and negative frequencies are not reported.

Response: We thank this reviewer for pointing this out. The TD-DFT calculations for **3a** has also been done and the simulated spectrum are qualitatively in accordance with the absolute configuration determined by XRD.

We have rephrased the description according to this suggestion and the revised version are shown below:

“Conformational searches on the transition states and intermediates were initially performed using Grimme's programs xTB 6.3 and CREST 2.10.2. All the conformers within 3 kcal/mol were further optimized at the level of B3LYP-D3/CPCM/Def2SVP level of theory.

The negative frequencies have been updated to the revised SI according to the suggestions.”

The HPLC should be revised. In particular: Why has rac-**3q** two enantiomers and four peaks? Why have Rac-**3ai**, **3aj**, **3ak**, **3al** two diastereoisomers and only two peaks? Moreover, it is apparently strange that a diastereomerization (i.g. Rac-**3ai**) yields 50% of each diastereoisomer.

Response: We thank this reviewer for raising this issue. Yes, the rac-**3q** only have two peaks on HPLC, referring to the two enantiomers, and other two peaks are corresponding two enantiomers of substrate **1a** (see below). Since the polarity of the product rac-**3q** and the substrate **1a** are the same, the mixture (bearing unreacted starting material **1a**) could not be separated by column chromatography, therefore there are four peaks in HPLC spectrum. In the synthesis of atropisomers with diaxes, substrate **1a** was not completely consumed, resulting in low yield (50%), and the low yield was not due to diastereomerization.

Peak Results

	Retention Time (min)	Int Type	Width (sec)	Area ($\mu\text{V}\cdot\text{sec}$)	Height (μV)	% Area
1	5.881	BB	58.000	546197	58157	50.74
2	9.346	BB	55.000	530168	37725	49.26

Syn/Anti disposition of Methoxy group could give a significantly different ratio. DFT calculations might help explain it.

Response: We thank this reviewer for pointing this out. The various orientations of the methoxy-naphthyl group correspond to distinct configurations of diastereomers. Additionally, we have also taken into account the relative orientation of the two

methoxy groups within the same configuration. Following calculations, we discovered that conformers featuring an anti-disposition of the methoxy group are marginally higher in energy (less than 1% in Boltzmann distribution) than those with a syn-disposition, which are presented in the Supporting Information.

Figure S2. 3D geometry structures and Boltzmann distribution of conformers of **3ai**.

Please also consider the points below:

Introduction 1) Please Remove “And” at the beginning of the sentence “And transition-metal-catalyzed Dykat”

Response: We thank this reviewer for pointing this out. “And” has been removed.

Results 2) Regarding the sentence: “However, the Lewis acidic boronic complex **1a**-Pd, the intermediate after oxidative addition of **1a** to Pd followed by ligand exchange, allows the coordination of the hydroxide ligand to form a tetracoordinate boron species with an elongated C–B axis. The corresponding rotational barrier from **1a**-Pd to **1a**-Pd’ is significantly reduced to 16.7 kcal/mol (Fig. 2c), which makes the free rotation of the aryl group around C–B stereogenic axis feasible, and fully support our conjecture.” I didn’t find how they have calculated GSs of **1a**-Pd, **1a**-Pd’ and the relative TS. Please introduce in the text or in the SI a better discussion of it.

Response: We sincerely thank this reviewer for the suggestions. Within the reaction mechanism, **1a**-Pd serves as an intermediate prior to the coordination of -OH to boron. However, it is worth noting that the coordinated intermediate is notably higher in energy than **1a**-Pd. This process shares similarities with the energy landscape demonstrated in Fig. 3, whereby the full model of ligand was utilized.

3) In Figure 2 the chemdraw structure of **1a**-Pd not shows the coordination between Boron and OH.

Response: We thank this reviewer for pointing this out. The coordination between Boron and OH in key the intermediate in chemdraw structure has been indicated in Figure 1d, which is highlighted by blue background (please see below as well as in Figure 1 in main text). In Figure 2, the coordination between Boron and OH has been shown by TS-Pd, which was listed on the arrow of eq. Fig 2c. To make it clear, we also revised the chemdraw structure of **1a**-Pd and **1a**-Pd'. Please see our revised manuscript.

4) Please change “Finally, reactions with chiral 3-bromo-2,1-azaborine **1a** were carried out,” with “Finally, reactions with enantioenriched 3-bromo-2,1-**1a** were carried out,”

Response: We thank this reviewer for pointing this out. This sentence has been revised, please see our revised manuscript.

5) Please Change “(**3a** and (-)-**3a**)” with “(**3a** and M-**3a**)”

Response: We thank this reviewer for pointing this out. Because the mechanistic experiment has been modified, this description (**3a** and (-)-**3a**) has been deleted. Please see our revised manuscript.

6) To reinforce their conjecture, they could do a kinetic experiment starting with **3a** enantiopure with the two different ligands.

Response: We thank this reviewer for pointing this out. Kinetic experiments of have been carried, please see our revised Fig. 3b.

7) In Figure 4, the compounds **3ai**, **3aj**, **3ak**, **3al** are diastereoisomers, they should insert the d.r.

Response: We thank this reviewer for pointing this out. The dr values have been inserted, please see our revised manuscript.

Supporting Information 1) ^{19}F NMR is missing. The CH_2 of Bn is an AB system in chiral environment. That means they are two different diastereotopic protons. Sometime it was written as quartet, instead of two distinct doublets.

Response: We deeply thank this reviewer for pointing this out. Per the request, ^{19}F NMR has been added, and the CH_2 of Bn has been revised. Please see our revised Supporting Information.

2) The ^{11}B NMR spectra could be better done in a quartz tube to remove or reduce the broaden band at 0 due to the borosilicates of glass and/or of probe material.

Response: We sincerely thank this reviewer for pointing this out. We are sorry that the ^{11}B NMR spectra could not be done because this project started in 2020 and most of the compounds have been decomposed or lost. The ^{11}B NMR spectra of the newly

synthesized compounds **1m**, **1n**, **3ai** and **3aj** were tested in a quartz tube, please see our revised Supporting Information.

Reviewer #2

The manuscript from Song, Yu et al. entitled “Construction of C-B axial chirality via dynamic kinetic asymmetric cross-coupling mediated by tetracoordinate boron” describes the enantioselective synthesis of azaborines bearing a C-B chiral axis, using a Pd-catalyzed dynamic kinetic asymmetric transformation (DyKAT). The methodology was extended to the concomitant formation of both a C-B and C-C chiral axis. The corresponding cyclic organoborons are usually obtained with excellent yields and enantioselectivities. According to the authors, and supported by DFT calculation, the formation of a tetracoordinate boron intermediate is the key of the success of this transformation. The preparation of atropisomers bearing a C-B chiral axis is a very challenging task, with only few examples described (to my knowledge 6 including 3 asymmetric synthesis). In the context of C-B atropisomers synthesis, the authors present an original approach based on DyKAT, a strategy that has been recently described by Stoltz and Lasseletta for the atroposelective synthesis of heterobiaryls. This is very interesting to apply this concept to the formation of C-B chiral axis, and it is a significance work in the field of atropisomers.

Response: We deeply thank this reviewer for the favorable comments on our work, we really appreciate it.

I have however, some concerns regarding the way the authors present this work. Indeed, the whole hypothesis relies on the formation of a tetracoordinated boron atom that apparently allows the DyKAT on these particular substrates. This is only supported by DFT calculations. I am not fully convinced by this hypothesis, and I think additional experimental data are required to confirm or not this hypothesis.

First, only substrates with a 2-methoxy group on the naphthyl moiety are used. DFT calculations suggest that the OMe is involved in the DyKAT. What about other substituents? Some substrates with a non-coordinating group should be tested to strengthen the calculations.

Response: We are grateful to the reviewer for bringing up this question. Based on our DFT calculations, the isomerization process takes place upon the coordination of -OH to the boron center. In addition, substrates with OEt or SEt group were tested under the standard conditions. A slightly lower ee value (89% ee) was observed by changing the OMe group to an OEt group with larger steric hindrance. Low enantioselectivity was obtained by using the SEt group with larger steric hindrance and stronger coordination. The results indicated that steric hindrance plays an important role in this DyKAT process.

One can argue that a coordination could also occur between the palladium and the OMe group. Has this hypothesis been rejected? It would be interesting to calculate the rotational barrier involving such coordination. What happens when a bulkier OR group is used, that could prevent the coordination?

Response: We would also like to express our gratitude to the reviewer for the suggestions. Although we were unable to locate the corresponding transition state, our scan calculation indicated that rotation along the C-B bond in the OMe coordinated intermediate would be incredibly challenging with an energy barrier of over 45 kcal/mol. We infer from this result that the rotation of the C-B bond may not occur via the coordination of OMe to Pd.

For the scope of the reaction, other groups that methoxy should be evaluated in the transformation (ie. OEt, OPh, SMe, ...).

Response: We thank this reviewer for pointing this out. Other groups (OEt and SEt) were evaluated the transformations. OEt group was tolerated well, but lower ee value

was obtained by using SEt group. The corresponding substrate with OPh group was not obtained.

In Figure 3a, the yields and ee of product **3a**, and recovered SM **1a** are plotted versus time. The authors should indicate in the text the initial ee for this experiment (racemic?). Since they use SM **1a** with 33% ee in Figure 3b and c, it could be a bit confusing.

Response: We thank this reviewer for pointing this out. The substrate **1a** in Fig 3a is racemic, which is indicated in Fig 3a. Please see revised manuscript.

In Figure 3a, the authors suggest a kinetic resolution and then a dynamic kinetic resolution. How do they draw this conclusion? When does the dynamic kinetic resolution started on Fig 3a? At 12h hours, the ee of **1a** is about 63% ee, then it drops down to ~33% ee, which means that **1a** is racemized. First, I don't understand why the ee of SM **1a** is about 33% at the end of the reaction.

Response: We thank this reviewer for pointing this out. We carried out mechanism experiments again. In Fig. 3a and 3b, the results shown that two enantiomers of rac-**1a** were consumed together and one of the enantiomers was decreased more rapidly, suggesting a kinetic resolution (KR) process. Please see our revised manuscript.

Second, how **1a** is racemized? The authors shows in Fig 3b that SM **1a** is not racemized with NaHCO_3 in toluene/water. Does it racemize under standard condition (with NaHCO_3 , Pd/ligand, Tol/ H_2O) without the aryl BF_3K ? Does the oxidative addition of the palladium reversible? (see *J. Organomet. Chem.* **2020**, 121571). Additional experiments should be done, and more explanations provided to help the reader understand.

Response: We thank this reviewer for pointing this out. We performed racemization experiment of chiral-**1a** under standard conditions without aryl BF_3K (Fig. 3c), and no obvious racemization of chiral-**1a** was found, excluding a dynamic kinetic resolution (DKR) pathway. The result also could rule out the reversible oxidative addition of the palladium.

Otherwise, the experimental part is well written with all the requested data. However, I didn't find the atom coordinates for IM1. Appropriate references are also included.

Response: We thank this reviewer for pointing this out. We have updated the atom coordinates for IM1 in the revised SI.

Other comments:

1) For the synthesis of compounds with two chiral axes, Li_2CO_3 was used, but this base hasn't been tested in Fig 2d. What is the results with Li_2CO_3 on the model substrate?

Response: We thank this reviewer for pointing this out. 92% yield and 93% ee were obtained with Li_2CO_3 as base, and the results have been added to Fig. 2d. Please see our revised manuscript.

2) Fig 3c. What is the conclusion of this experiment?

Response: We thank this reviewer for pointing this out. We have updated the mechanism experiments. Previous experiment in Fig. 3c has been deleted.

3) The absolute configuration was determined by X-ray of **3a**. Is it consistent with the DFT calculations?

Response: We thank this reviewer for pointing this out. The TD-DFT calculations for **3a** has also been done and the simulated spectrum are qualitatively in accordance with the absolute configuration determined by XRD.

4) Fig 4: compounds **3n**, **3p**, **3r**, **3ag**, **3ai** with yields below 50%. Since yields are lower than 50%, is it a DyKAT process or a kinetic resolution?

Response: We thank this reviewer for pointing this out. Incomplete conversion of substrate leads to low yields. And mechanistic experiments show that there is a kinetic resolution process in this DyKAT process. Please see our revised Fig. 3.

Reviewer #3

In this manuscript, Song, Yu and co-workers report a novel dynamic kinetic asymmetric transformation (DyKAT), accomplished via a tetracoordinate boron-directed diastereomeric intermediate equilibrium process that differs from previous chelation-directed DyKAT of racemic heterobiaryls. This equilibrium process is well designed, innovative and effective, and it is also supported by DFT calculations and control experiments. Through this methodology, a variety of C-B axially chiral compounds

were prepared in high enantioselectivity. More interestingly, atropisomers with C-B and C-C diaxes were also prepared in high diastereoselectivity and enantioselectivity. This methodology has also been applied to complex molecules. The manuscript and supplementary information were well written and prepared. For these reasons, I believe that this work is suitable for publication in Nature Communications after consideration of the following comments:

Response: We sincerely thank this reviewer for the favorable comments on our work, we really appreciate it.

1. DFT calculations well supported the design of this interesting dynamic kinetic asymmetric transformations and helped determine the absolute configuration for adjacent C-B and C-C diaxially chiral compound. The energy barrier of rotation around the C-B bond in **1a**-Pd was calculated to be much lower than that in the substrate, suggesting the transformation between two different configurations is enabled by the Pd catalyst. This rotation relies on the coordination of hydroxide ligand. I was wondering whether the ligand exchange from bromide to hydroxide is thermodynamically feasible. Considering the significant role of the hydroxide, I suggest that the authors should try their best to search for more evidence and details from the literature or perform further calculations to support the possibility of the formation of Pd-OH complex.

Response: We thank this reviewer for the suggestions. The ligand exchange from bromide to hydroxide was calculated to be exothermic by 10 kcal/mol. The similar mechanism has been reported by Maseras et al. (*ChemCatChem* **2014**, *6*, 3132–3138).

2. In abstract, the authors should use the more specific “diastereo- and enantio-control” instead of “stereo-control”.

Response: We thank this reviewer for pointing this out. Per the request, “diastereo- and enantio-control” has been used.

3. Fig. 2d caption, the authors should add NMR Internal standard.

Response: We thank this reviewer for pointing this out. NMR Internal has been added.

4. Page 5, line 18, “Pd₂(dba)₂” should be replaced by “Pd₂(dba)₃”.

Response: We thank this reviewer for pointing this out. “Pd₂(dba)₂” has been replaced by “Pd₂(dba)₃”.

5. Page 7, line 1, “stereoselectivities” should be replaced by “enantioselectivity”.

Response: We thank this reviewer for pointing this out. “stereoselectivities” has been changed into “enantioselectivity”.

6. Why is the yield of product **3ag** so low? Is kinetic resolution involved here?

Response: We thank this reviewer for pointing this out. Some amount of substrate remained, but the ee value of substrate was low. A kinetic resolution process is involved in the transformation.

7. How about using alkyl trifluoroborates as the substrate? Some commentary should be added.

Response: We thank this reviewer for pointing this out. The transformation was not compatible with alkyl trifluoroborates, and no corresponding product was obtained.

8. In supplementary information, the authors should add dr values of compounds **4b** and **4c**.

Response: We thank this reviewer for pointing this out. The dr values of compounds **4b** and **4c** have been added.

Reviewers' Comments:

Reviewer #1:

Remarks to the Author:

Since the first review the DyKAT process has been better described.

However, some points have not been resolved. Specifically, the ECD spectrum of 3ak was cut at 250 nm in n-hexane and only 30 nstates were considered. In the calculation with 70 nstate the large negative band at 210-220 nm of the naphthyls is evident. Authors should collect experimental ECD spectra in different solvents not to cut this band (e.g. in acetonitrile). This band is critical for these types of compounds, because it is the band related to these naphthyl chiral axes. If the authors want to be sure of both chiral axes, they could also perform the NOESY-1H NMR experiment to check if the two OMe are on the same or opposite side.

For the ECD and NOE experiment, they could see:

J. Org. Chem. 2013, 78, 8, 3709–3719 doi.org/10.1021/jo400200v

J. Org. Chem. 2017, 82, 13, 6874–6885 doi.org/10.1021/acs.joc.7b01010

Synlett 2018; 29(16): 2161-2166 DOI: 10.1055/s-0037-1609908

For a better understanding of figure S2, the authors should insert the descriptors to differentiate the geometries found (for example dihedral angles of substituents).

Not explication was given to HPLC of diastereomeric compounds. The racemic mixtures should give four peaks and not two. Are the two peaks shown enantiomers or diastereoisomers? They should assign the absolute configuration at both diastereoisomers and they need to check if they are reading an ee or a dr.

The 11B NMR have not been corrected, I let the editor to decide if they are good enough for publication.

Some othrs corrections:

Chiral descriptor should be Ra or Sa not only R o S.

Pag3. Add boron: After oxidative addition, and anion ligand exchange, benefiting from the "boron" Lewis acidity

Pag3. The C-B bond in IM1 is free to rotate with a

Pag3. In the meanwhile ◊ Meanwhile

Fig. 3 missing superscript symbol of ΔG^\ddagger

Pag5. Enantiopure 1a (37% ee) ◊ enantioenriched 1a (37% ee)

Pag6. The authors could better explain that the tetracoordinate boron species is also chiral to boron tetrahedral atom.

Fig.1 chiral Pd catalysis is above the arrow

Reviewer #2:

Remarks to the Author:

The authors performed all the additional experiments and corrections asked. I recommend publication in Nature Communications

Reviewer #3:

Remarks to the Author:

I recommend the publication of the revised manuscript.

Point-by-point Response to Reviewer(s)' Comments

Reviewer(s)' Comments to Author:

Reviewer #1 (Remarks to the Author):

Since the first review the DyKAT process has been better described. However, some points have not been resolved. Specifically, the ECD spectrum of **3ak** was cut at 250 nm in n-hexane and only 30 nstates were considered. In the calculation with 70 nstate the large negative band at 210-220 nm of the naphthyls is evident. Authors should collect experimental ECD spectra in different solvents not to cut this band (e.g. in acetonitrile). This band is critical for these types of compounds, because it is the band related to these naphthyl chiral axes. If the authors want to be sure of both chiral axes, they could also perform the NOESY-1H NMR experiment to check if the two OMe are on the same or opposite side.

For the ECD and NOE experiment, they could see:

J. Org. Chem. 2013, 78, 8, 3709–3719 doi.org/10.1021/jo400200v

J. Org. Chem. 2017, 82, 13, 6874–6885 doi.org/10.1021/acs.joc.7b01010

Synlett 2018; 29(16): 2161-2166 DOI: 10.1055/s-0037-1609908

For a better understanding of figure S2, the authors should insert the descriptors to differentiate the geometries found (for example dihedral angles of substituents).

Response: We express our gratitude to the reviewer for pointing this issue and providing valuable solutions. For **3ak**, we have reconducted the ECD experiments using acetonitrile as solvent and collected the experimental data ranging from 210 nm to 400 nm. The TD-DFT calculations have been performed at the level of ω B97X-D/SMD(acetonitrile)/Def2TZVP theory with 70 transitions (NSATES=70) as recommended by this reviewer. The comparison between the experimental ECD spectra and the simulated ECD spectra of different configurations supports **3ak-anti** as the major product featuring two OMe on opposite side.

In the meanwhile, Two-dimensional NMR identified two OMe groups are *opposite side* referring to the borazaronaphthalene ring.

Analysis: 1) The single aromatic peak was attributed as 17-CH; 2) No matter the *cis* or *trans*, the NOE correlation with 17-CH is only with 9-CH₃; 3) The aromatic double-peak hydrogens that have possible correlations with two OMe groups include 6-CH, 10-CH, 7-CH and 14-CH; 4) The HMBC experiment made the 6-CH and 10-CH confirmed (Figure S2, S3 and S4); 5) between 6-CH and 10-CH, the only one has correlation with benzylic H (C1-H) should be 7-CH, thus confirming the 7-CH and 14-CH; 6) The correlation between 7-CH and 9-CH₃, as well as the correlation between 14-CH and 5-CH₃ indicated the two OMe groups are *opposite side*, referring to the borazaronaphthalene ring.

Figure S6. NOE spectrum-1 of 3ak in toluene-d₈.

Figure S7. HMBC spectrum-1 of 3ak in toluene-d₈.

Figure S8. HMBC spectrum-2 of 3ak in toluene-d8.

Figure S9. NOE spectrum-2 of **3ak** in toluene-d8.

The references for determining absolute configuration using ECD and NOESY experiments have been cited in the manuscript. Combined above experimental observation and computational results, we assign major product **3ak**. Based on the above results, we have revised the corresponding Lewis structure and notion about the absolute configuration of **3ak** for both manuscript and SI.

Not explication was given to HPLC of diastereomeric compounds. The racemic mixtures should give four peaks and not two. Are the two peaks shown enantiomers or diastereoisomers? They should assign the absolute configuration at both diastereoisomers and they need to check if they are reading an ee or a dr.

Response: We thank this reviewer for pointing this out. According to NMR analysis, the synthesis of racemates for atropisomers with diaxes has good diastereoselectivities. The two peaks of racemates are shown enantiomers.

The ^{11}B NMR have not been corrected, I let the editor to decide if they are good enough for publication.

Response: We appreciate this reviewer's understanding on this issue. We are sorry that the ^{11}B NMR spectra could not be redetected because this project started in 2020 and most of the compounds have been decomposed or lost.

Some others corrections:

Chiral descriptor should be Ra or Sa not only R o S.

Response: We thank this reviewer for pointing this out. R or S have been changed to Ra or Sa. Please see our revised manuscript and SI.

Pag3. Add boron: After oxidative addition, and anion ligand exchange, benefiting from the "boron" Lewis acidity.

Response: We thank this reviewer for pointing this out. The "boron" has been added. Please see our revised manuscript.

Pag3. The C-B bond in IM1 is free to rotate with a ...

Response: We thank this reviewer for pointing this out. This sentence has been revised, please see our revised manuscript.

Pag3. In the meanwhile □ Meanwhile

Response: We thank this reviewer for pointing this out. "In the meanwhile" has been changed to "Meanwhile". Please see our revised manuscript.

Fig. 3 missing superscript symbol of □G#

Response: We thank this reviewer for pointing this out. Although we saw "□G#" in the

system, we guess that this reviewer requires the addition of superscript of ‡. ΔG^\ddagger is usually used to denote activation barrier, while ΔG here was used to indicate the relative free energies for both intermediate and transition state with respect to **IM0**.

Pag5. Enantiopure 1a (37% ee) □ enantioenriched 1a (37% ee)

Response: We thank this reviewer for pointing this out. “Enantiopure 1a (37% ee)” has been changed to “Enantioenriched 1a (37% ee)”. Please see our revised manuscript.

Pag6. The authors could better explain that the tetracoordinate boron species is also chiral to boron tetrahedral atom.

Response: We thank this reviewer for pointing this out. The “form a tetracoordinate boron species” has been changed to “form a chiral tetracoordinate boron species”. Please see our revised manuscript.

Fig.1 chiral Pd catalysis is above the arrow

Response: We thank this reviewer for pointing this out. This issue has been revised. Please see our revised manuscript.

Reviewers' Comments:

Reviewer #3:

Remarks to the Author:

I have examined the comments from the reviewer and corresponding response from the authors, and I believe the authors have completely solved the issues, so I recommend the publication of the revised manuscript.

Point-by-Point Response to the reviewers' comments

REVIEWER COMMENTS

Reviewer #3 (Remarks to the Author):

I have examined the comments from the reviewer and corresponding response from the authors, and I believe the authors have completely solved the issues, so I recommend the publication of the revised manuscript.

Our response: We sincerely thank this reviewer for his/her recommending our manuscript publication in *Nature Communications*. Meanwhile, we also would like to show our gratitude to this reviewer for his/her contribution to improve the quality of our manuscript.